# Determinants of COVID-19 Vaccine Uptake among Female Foreign Domestic Workers in Hong Kong: A Cross-Sectional Quantitative Survey

**DOI:** 10.3390/ijerph19105945

**Published:** 2022-05-13

**Authors:** Timothy S. Sumerlin, Jean H. Kim, Zixin Wang, Alvin Yik-Kiu Hui, Roger Y. Chung

**Affiliations:** 1School of Public Health and Primary Care, The Chinese University of Hong Kong, Hong Kong 999077, China; timsumerlin@link.cuhk.edu.hk (T.S.S.); jhkim@cuhk.edu.hk (J.H.K.); wangzx@cuhk.edu.hk (Z.W.); alvinhui@cuhk.edu.hk (A.Y.-K.H.); 2Institute of Health Equity, The Chinese University of Hong Kong, Hong Kong 999077, China; 3Centre for Bioethics, The Chinese University of Hong Kong, Hong Kong 999077, China

**Keywords:** COVID-19, vaccination, foreign domestic workers, China

## Abstract

Globally, minority groups and non-citizens may not be sufficiently included in the COVID-19 vaccine coverage. This study seeks to understand determinants of vaccine uptake among female foreign domestic workers (FDWs) in Hong Kong. We conducted a cross-sectional study of female FDWs (*n* = 581) from June to August 2021. Respondents completed an online survey obtaining sociodemographic, employment, and health status information. Based upon the socio-ecological model, we obtained individual, interpersonal, and socio-structural factors that may be associated with COVID-19 vaccine uptake. Multivariable logistic regression analysis was used to examine factors associated with having received at least one dose of a COVID-19 vaccine. At the individual level, agreeing that taking COVID-19 vaccines can contribute to COVID-19 control in Hong Kong (OR 6.11, 95% CI 2.27–16.43) was associated with increased vaccine uptake, while being worried of severe side-effects from vaccination (OR 0.29, 95% CI 0.16–0.55) was associated with decreased uptake. At the interpersonal level, those being encouraged by their employer (OR 2.05, 95% CI 1.06–3.95) and family members (OR 2.27, 95% CI 1.17–4.38) were more likely to be vaccinated, while at the socio-structural level, believing vaccination would violate religious beliefs (OR 0.19, 95% CI 0.06–0.65) was associated with decreased uptake. The government can formulate a multi-level approach according to our findings to target the remaining unvaccinated FDW population.

## 1. Introduction

Coronavirus disease 2019 (COVID-19) continues to surge globally, since first being identified in late 2019, with 517 million confirmed cases and over 6.2 million deaths as of May 2022 [1]. Over 11.3 billion vaccine doses have been administered globally as of May 2022 [1]. Due to the spread of the Omicron variant, and other potential new and more infectious variants, international health organizations are recommending all eligible people to receive a COVID-19 vaccine. Understanding the determinants of COVID-19 vaccine uptake is crucial in achieving high vaccine coverage.

Hong Kong, a special administrative region of China, has enacted stringent infection control measures which included border closures to non-residents, compulsory 3-week hotel quarantines for international arrivals, mask requirements in public areas, and large-scale community testing when a local case is confirmed [2]. Despite being a densely populated city of 7.4 million [3], just over 12,000 confirmed cases and 213 deaths were reported at the end of 2021 [4]. However, a surge of cases beginning in early 2022 caused the confirmed case count to reach over 1.3 million as of May 2022 [2]. In May 2021, the Hong Kong government’s “vaccination for all” campaign began offering two types of COVID-19 vaccines to all adults [5]. The first, BNT162b2 (Comirnaty), produced by Fosun-BioNTech, is an mRNA vaccine, while the second, CoronaVac, produced by Sinovac-Biotech, is an inactivated virus vaccine [6]. As of May 2022, 91.4% of those aged 3 or older have taken at least one dose of a COVID-19 vaccine [7].

Vaccine hesitancy and vaccine uptake have been previously studied in the general Hong Kong population. A serial cross-sectional survey of Cantonese-speaking working adults found that a decrease in willingness to be vaccinated was associated with an increase in concern for vaccine safety [8]. Another serial cross-sectional study of Cantonese-speaking adults aged 18 and above found prior to the “vaccination for all” campaign, vaccine hesitancy was highest in young adults (aged 18–34), while after the launch of the campaign hesitancy was highest in older adults (aged ≥ 65) [9]. Furthermore, higher vaccine uptake was positively associated with middle-age (aged 35–59), higher educational attainment, higher confidence in COVID-19 vaccines, and greater trust in their own ability to prevent infection [9]. However, both studies only included Cantonese-speaking permanent residents of Hong Kong, excluding ethnic minority populations and those without permanent residency status. A single study on determinants of COVID-19 vaccine uptake in the South Asian ethnic minority adult population of Hong Kong was conducted in May 2021 and found a relatively low vaccine uptake rate of 33.1% among its 245 respondents [10]. Those with positive attitudes toward vaccination, perceived support from significant others, and perceived higher behavioral control to receive vaccination were positively associated with vaccine uptake, while higher exposure to information about deaths and serious conditions caused by COVID-19 vaccination was associated with decreased vaccine uptake [10].

While Hong Kong’s population is predominately ethnically Chinese, female foreign domestic workers (FDWs), primarily from the Philippines and Indonesia, are Hong Kong’s largest group of ethnic minorities accounting for 5% of the total Hong Kong population (i.e., 373,384) in 2020 [11]. All FDWs must enter Hong Kong under the same working visa scheme in which they are required to repatriate within 2 weeks if their full-time contract is terminated [12]. This population is uniquely distinguished from other ethnic minority populations in Hong Kong in two key ways. First, their FDW status provides no path to permanent residency in Hong Kong, whereas all other migrants have a path to permanent residency [13]. Second, FDWs are required to live with their employers (“live-in” rule), typically characterized by small living quarters, a high intensity of interpersonal contact, and other vulnerabilities which have been previously documented [13,14]. Early in the COVID-19 pandemic, Marmot & Allen [15] warned how the pandemic would expose and amplify the inequalities in society, while others early in the pandemic specifically called for increased attention to international migrant workers [16]. One study conducted in May 2020 found the rate of probable anxiety among FDWs in Hong Kong to be at 25%, with lack of protective equipment, increased workload, and fears of being fired if infected by COVID-19 to be associated with probable anxiety [17]. Qualitative interviews conducted in 2020 among 15 FDWs in Hong Kong reported poor treatment from their employers, such as having excessive expectations for cleaning during the pandemic and unfair termination of job due to the pandemic [18]. News reports have documented how FDWs felt discriminated and unfairly scrutinized throughout the pandemic [19]. Furthermore, several government policies, such as required testing of all FDWs, have been criticized as FDWs were required to stand in long lines without social distancing to be tested, with many losing their only day off for the week [20]. Additionally, in May 2021, the government announced and then retracted a policy requiring the vaccination of all FDWs who wished to keep their jobs [21]. These circumstances have created a sense of uncertainty within the FDW population. Furthermore, there is no available data on vaccine uptake among the FDW population in Hong Kong.

The socio-ecological model has been used to assess COVID-19 vaccine acceptance and uptake in several studies [10,22,23,24,25]. The socio-ecological model is a theoretical framework commonly used in social sciences and public health which considers both individual and societal factors contributing to health outcomes [26]. The socio-ecological model comprises sociodemographic factors, individual, interpersonal, and socio-structural factors [26]. Informed by the socio-ecological model, a scoping review including 50 studies assessed factors which influenced public attitudes towards COVID-19 vaccination globally and found contributing factors at each level [22]. At the sociodemographic level, factors associated with positive attitudes towards vaccination included female sex, age older than 25, higher educational attainment, high income status, chronic diseases, being employed, and being married with children [22]. At the individual level, those who trusted the government, healthcare system, and companies producing the vaccines had more positive attitudes towards vaccination, while those with conspiracy theory beliefs and those with religious reasons for refusing the vaccine were more vaccine hesitant [22]. At the interpersonal level, positive messaging from healthcare professionals, family, and friends were associated with positive attitudes towards vaccination, while one study found employer’s recommendation may also contribute to a positive vaccine attitude [22,27]. Finally, at the socio-structural level, both traditional news media and social media were contributing factors regarding vaccination attitudes [22]. Addressing determinants at multiple levels may increase the likelihood of changing health behaviors. To the best of our knowledge, there are no studies examining the determinants of vaccine uptake among FDWs anywhere in the world. As there are over 8.5 million female FDW globally [28], the findings of this study may help to improve vaccine uptake for FDWs in Hong Kong and in other countries.

## 2. Materials and Methods

Data was collected between 16 June and 29 August 2021. Non-probability sampling was used to reach both Filipino and Indonesian FDWs through various social media platforms due to an inability to recruit respondents in public places because of COVID-19 social distancing restrictions. Key volunteers, leaders within the FDW community who are also FDWs, helped disseminate the survey through FDW groups on these platforms. Indonesian FDWs were recruited through a WhatsApp group where the key volunteer messaged each group member, while Filipino FDWs were recruited in a Facebook group where the invitation to participate in our study was posted in the group thread. Inclusion criteria were female FDWs from the Philippines or Indonesia, of any age, currently working in Hong Kong as an FDW for at least one year and had the ability to complete a self-administered questionnaire in English or Bahasa Indonesia. The Bahasa Indonesia version was translated from the original English version and cross-checked by a second native Bahasa Indonesia speaker. Both languages of the survey were pilot tested using cognitive debriefing with FDWs prior to mass distribution of survey to ensure understanding and clarity, and changes were made where needed. Using a population size of 373,384, a margin of error of 5%, a confidence interval of 95%, and a 50% response distribution, the recommended sample size for this study was at least 384 participants. Written informed consent was obtained before completion of the survey and ethics approval was granted by the Survey and Behavioural Research Ethics Committee (SBREC) of The Chinese University of Hong Kong (Reference No. SBRE(R)-21-018).

### 2.1. Measures

Several questions related to respondents’ socio-demographic background, employment conditions, and health were asked to control for potential confounding factors. Respondents were asked about their ethnicity, age, educational attainment, and marital status. Questions related to participants’ employment conditions included employer’s home size, as a proxy for employer’s socioeconomic status, and past month average working hours. The EQ-5D-5L Visual Analog Scale was used to measure self-rated health, asking respondents to rate their health today on a scale from 0 (worst health) to 100 (best health). Respondents were also asked whether they have previously been diagnosed with any chronic health condition. To assess the main outcome, COVID-19 vaccine uptake, participants were asked whether they had taken at least one dose of any COVID-19 vaccine (yes or no), number of doses (1 or 2), and type of vaccine (CoronaVac or Comirnaty). To control for the increase in vaccine uptake rate over the data collection period, the first day of data collection was counted as “day 0” and each day counted from there to note when the survey was completed. This variable was included as a continuous variable and ranged from 0 to 71.

Informed by the socio-ecological model, questions related to individual, interpersonal, and socio-structural factors were developed. At the individual level, participants provided their level of agreement (5-point Likert scale, strongly agree to strongly disagree) to four positive belief statements (1) vaccination is highly effective in protecting you from COVID-19, (2) vaccination is highly effective in protecting those around you from COVID-19, (3) vaccination facilitates resumption of cross-border travel, and (4) vaccination contributes to the control of COVID-19 in Hong Kong and one negative statement (1) I am concerned about of having severe side-effects. Interpersonal level factors included respondent’s social networks, whether their employer forced, encouraged, discouraged, or was neutral towards their vaccination, whether their family members encouraged vaccination, and whether their FDW friends encouraged vaccination. Whether they knew any FDW who had serious side effects after vaccination was also asked. Questions related to the source (social media, employer, friends, family, news) and frequency (almost never, seldom, sometimes, always) of obtaining COVID-19 vaccine-related news were asked. Whether overall information obtained about COVID-19 was positive, negative, or equally positive and negative was asked. Socio-structural questions included two questions related government policy to incentivize vaccination. These were, “I was motivated to get the COVID-19 vaccination to be” (1) exempt from the required COVID-19 swab testing, and (2) eligible for the 100k HKD lucky draw. To assess the influence of religious beliefs, participants were asked whether they believed COVID-19 vaccination would violate their religious beliefs. For analysis, variables measured on the 5-point Likert scale were dichotomized to disagree/not sure vs. agree.

### 2.2. Data Management

News source and frequency were reduced to a single item using Principal Component Analysis (PCA) and reliability analysis (RA). The Kaiser-Meyer-Olkin Measure was 0.826 (above 0.6) and Bartlett’s Test was statistically significant at *p*-value < 0.001 (<0.05), both indicating the items were appropriate for PCA. One component had an Eigenvalue above 1 at 2.881, indicating the five items could fit into one component. This was confirmed by the Scree plot suggesting one component. RA confirmed this with the five items producing a Cronbach’s Alpha of 0.809. This news information scale score ranged from 5 to 20 and we categorized the outcome based on inter-quartile range where ≤13 = low, 14–18 = average, and ≥19 = high.

### 2.3. Statistical Analysis

Descriptive statistics with percentage and frequency were calculated. When appropriate, mean and standard deviation were used. PCA and RA were used to see whether the five items related to the source and frequency of COVID-19 vaccine news could be reduced into a unidimensional scale with a single summative score. Binary logistic regression was used to explore the variables associated with vaccine uptake (taken at least one dose of COVID-19 vaccine) and the unadjusted odds ratios (OR) and 95% confidence intervals (CI) were calculated. All variables which had a *p*-value < 0.1 in the univariable analysis were included into a final multivariable model. All statistical analyses were performed using SPSS version 24 (IBM Corp., Armonk, NY, USA) [29].

## 3. Results

### 3.1. Descriptive Results

Table 1 describes the characteristics of the study population. A total of 581 female FDWs completed the survey, of which, 60% were Filipino and 40% Indonesian. Most participants were between the age of 35 to 44 (53%), 49% had up to a secondary school education, and nearly half were married (47%). Almost one-third claimed their employer’s home size to be ≤599 sq. ft., while 26% reported a home size ≥ 1000 sq. ft. The mean daily working hours was 13.0 h (6 days a week). Chronic health conditions were reported by 5.7%. Most participants reported receiving at least one dose of a COVID-19 vaccine (79%), with 2% of this group reporting serious side effects. Most vaccinated respondents already received their second dose (87.7%) with 71% of vaccinated respondents choosing the Comirnaty vaccine. Among unvaccinated participants, 39% said they were likely to receive a COVID-19 vaccine within the next 6 months. Receiving a COVID-19 vaccine for personal health was the most important factor for vaccine uptake (43%), while concern over serious side effects from the vaccine was the most important reason for not receiving a vaccine (46%).

Perspectives related to COVID-19 vaccination are found in Table 2. Positive attitudes towards vaccination received high agreement, ranging from 82–89% agreement. Among social networks, those reporting encouragement from employers, being forced by employers, and encouragement from family members to be vaccinated all reported 88% vaccination rates. Respondents were more motivated to be vaccinated to avoid repeated testing than by the potential to receive a cash prize in a lucky draw.

The sources of COVID-19 vaccine-related information are shown in Table 3. Social media was reported to be the most frequently used source for vaccine-related news, while traditional news was the least used. Seeing news with positive coverage of COVID-19 vaccines saw the highest percentage of vaccine uptake at 89%.

### 3.2. Univariable Logistic Regression

Unadjusted odds ratios for vaccine uptake are shown in Table 4. Among demographic variables, being middle-aged, unmarried, and those whose employer’s home was larger were more likely to be vaccinated, while those working more hours per day were less likely to be vaccinated. All attitudinal/perception variables related to COVID-19 vaccination at the individual, interpersonal, and socio-structural level were statistically significantly associated with vaccine uptake. Additionally, obtaining COVID-19 vaccine-related information at a high frequency and reporting that vaccine-related news seen was positive were significantly more likely to be vaccinated (*p* < 0.05).

### 3.3. Final Multivariable Model

The final model of adjusted odds ratios is shown in Table 4. Middle-aged participants, those who have never been married, those living in a larger home, and respondents with less working hours were more likely to have been vaccinated. Among individual level factors, believing vaccination can contribute to the control of COVID-19 locally (OR 6.11, 95% CI 2.27–16.43) increased vaccine uptake likelihood, while having concerns of severe side-effects from the vaccine (OR 0.29, 95% CI 0.16–0.55) significantly decreased vaccine uptake. At the interpersonal level, receiving encouragement from employer (OR 2.05, 95% CI 1.06–3.95) and family members (OR 2.27, 95% CI 1.17–4.38) increased vaccine uptake. At the socio-structural level, perceiving vaccine uptake would violate one’s religious beliefs (OR 0.19, 95% CI 0.06–0.65) decreased vaccine uptake.

## 4. Discussion

To the best of our knowledge, this was the first study to assess COVID-19 vaccine uptake among the female FDW population, a vulnerable population in Hong Kong who are set apart from other ethnic minority groups in that they cannot seek permanent residency. Among our study sample, vaccine uptake was relatively high with 79% having taken at least one dose of a COVID-19 vaccine. For reference, at the end of our data collection period, August 2021, the general Hong Kong adult population’s vaccination rate was 55%, while a convenience sample of the South Asian ethnic minority population in Hong Kong reported a 33.1% vaccination rate, by the end of May 2021 [10,30]. Additionally, the vaccine uptake rate did not significantly differ between the Filipino and Indonesian FDWs in our study sample. Using the socio-ecological model, we found significant indicators for vaccine uptake at the individual, interpersonal, and socio-structural levels.

The relatively high vaccination rate among this sample may be attributed to several factors. First, it appears the government’s policies towards vaccinating FDWs did have an effect on vaccine uptake. While there is currently no vaccine mandate for FDWs already residing in Hong Kong, the implementation and then retraction of a vaccine mandate by the government along with two rounds of required COVID-19 testing of all FDWs may have inadvertently increased the desire for vaccine uptake [21]. In fact, 68% of our sample reported they were motivated to be vaccinated to be exempt from required COVID-19 testing. A study in six European countries found varying effectiveness of mandatory COVID-19 certification (vaccination, negative test, proof of recovery) on vaccine uptake, with those younger than 30 having the highest increase in vaccine uptake post certification implementation [31]. This differs from our sample which found the highest vaccine uptake among 35–44 years old, even though this could be due to different societal factors. Second, in contrast to past outbreaks where perceptions of susceptibility and illness severity decreased over time for diseases such as SARS, H1N1, and H5N1 avian influenza [32,33,34], the sense of vigilance and vulnerability to COVID-19 did not seem to deteriorate despite a relatively lower disease prevalence in Hong Kong, which can be made evident by the enduring mask wearing law, mandatory digital contact tracing that has only become more widespread, enduring border control, and enduring albeit less stringent local social distancing measures [2]. However, the 34% agreement of a lucky draw incentivizing vaccination points to prize incentives as less effective. In other words, policies affecting FDWs working opportunities in Hong Kong may be a more effective strategy moving forward. The risks and issues with financial incentives have been previously discussed, with researchers recommending incentives such as access to certain facilities given only to vaccinated people as more sustainable than cash incentives in the long term [35]. Third, there may be a strong desire for FDWs to return to their home country to visit family. At the time of data collection, FDWs would need to quarantine themselves when they arrive in their home country and when they return to Hong Kong, making a visit home unfeasible since the start of the pandemic. Qualitative interviews conducted in the summer of 2020 found FDWs in Hong Kong felt stressed by their inability to return home and fulfill familial duties [18]. Respondents agreed at high rates that vaccination can contribute to the control of COVID-19 in Hong Kong, which may be a proxy for the desire for resumption of returning to a normal life in which they can visit home again without quarantine.

Among the demographic characteristics, never married FDWs were more likely to take up vaccination, which differed from the South Asian ethnic minority group in Hong Kong, the general Hong Kong population, and the general findings of a scoping review where being married or cohabitating was positively associated with vaccine uptake/acceptance [8,10,22]. While those living with family members may feel a stronger sense of obligation to be vaccinated to protect their family, this effect seems to be missing among FDWs as they do not live with their immediate family. Further investigation is needed to determine why never married FDWs may be more likely to be vaccinated. Those living in a larger home, a proxy for employer’s socio-economic status, were more likely to have been vaccinated. Past literature in Hong Kong has pointed to lower acceptance of COVID-19 vaccines among those in occupations associated with lower socio-economic status [8]. It appears this may also influence the FDWs who are employed by this demographic, and therefore, it may be appropriate to consider the role of their employer in vaccine uptake when formulating policies and promotion of vaccination among FDWs.

At the individual level, believing vaccination can contribute to the control of COVID-19 in Hong Kong and concern of having severe side-effects from the vaccine were most strongly associated with increased and decreased vaccine uptake, respectively. Believing vaccination can contribute to the control of the virus in Hong Kong may be a proxy for returning to a more pre-pandemic way of life such as unrestricted gatherings on day off and visiting family in home country. Consideration of which type of gathering restrictions can be lifted for the FDW population based on their vaccination rate may be considered as an incentive to increase vaccine uptake. On the other hand, concern of severe side-effects is consistent with previous literature and requires more targeted education campaigns on vaccine safety towards the FDW community [8,10,22,36,37,38]. Among interpersonal factors, encouragement from employer or family to be vaccinated had the strongest associations with vaccine uptake. Association between encouragement from family and vaccine acceptance is consistent with the South Asian population in Hong Kong and globally [10,22]. While being forced by employer to be vaccinated having only a marginally significant association with vaccine uptake, our findings altogether may imply that encouragement rather than coercion in vaccine uptake should be of consideration when the government and employers communicate about vaccination with the FDW community. A multi-country survey early in the pandemic found that 48% of respondents would accept vaccination if their employer recommended it, though rates varied by country with China reporting the highest acceptance and Russia the lowest [27]. At the socio-structural level, while the belief that COVID-19 vaccination would violate one’s religious beliefs was significantly associated with decreased vaccine uptake, only 5% of our study sample agreed that it really violated their religious beliefs. Just nine people (1.5%) agreed with the statement and were unvaccinated, making this of less concern as a barrier to vaccine uptake at the population level. Nevertheless, increased collaboration and cooperation should be taken with religious leaders and the religious community to maximize vaccine uptake and address individuals’ concerns. One study from the Czech Republic found higher rates of vaccine hesitancy in those identifying as spiritual but with non-religious affiliation than those with religious affiliation [39]. More in-depth studies are needed to understand how exactly religious beliefs and religiosity may affect vaccine uptake and how to most effectively address these concerns.

Of those unvaccinated, concern about severe side effects were most common, as shown consistently in past literature [8,10,22,36,37,38]. The next most common reason was not having the chance but planning to be vaccinated. While the data for this study was collected starting 16 June 2021, FDWs were already included in the vaccine implementation plan since March 15 of the same year [40], giving them at least three months of access to the vaccine before data collection began. This may explain the significance of higher working hours being negatively associated with vaccine uptake. Past studies of FDWs have pointed to excessive working hours as detrimental to health as well as decreased likelihood of participating in health screening [14,41,42]. A study in Canada assessing barriers to healthcare access among FDWs found employers played a significant role in the ability for FDWs to access healthcare [43]. This further points to the positive role employers can play in the health of FDWs. Additionally, the government may consider opening mobile vaccination centers in the most popular Sunday gathering areas where FDWs spend their day off. This may improve vaccine uptake especially among those who reported not yet having adequate time to be vaccinated by increasing geographical accessibility to vaccination sites.

In summary, a more targeted approach is warranted to improve vaccine uptake in those who may still be hesitant. A health promotion campaign to educate FDWs on vaccine safety and efficacy should be tailored to the FDW community specifically rather than the general Hong Kong population. Additionally, the role that employers play in the rate of vaccine uptake among FDWs should be further considered. Finding approaches that facilitate the positive influence employers can have over FDW vaccine uptake should be promoted.

The study’s findings must be considered with some limitations. First, the cross-sectional study design does not allow for causal inference, while non-probability sampling limits our ability to provide a prevalence for vaccine uptake among the FDW population or generalize our findings to the FDW population. However, with the nature of this population being more elusive than the general population and the social-distancing restrictions from the COVID-19 pandemic, random sampling became very difficult. Additionally, our sample did not differ much from available population data. As of 2021, population data shows 57% of FDWs were Filipino and 41% Indonesian (our sample 60% and 40%, respectively), with 2% from other countries. Additionally, population data shows 35% aged 20–34, 45% aged 35–44, and 20% aged 45 or older, compared with 31%, 53%, and 16% in our sample, respectively [11]. There is no known available data on education level, income, or age based on ethnicity. Second, as our survey relies on self-reported information, our study is subject to recall bias and social-desirability bias, where respondents answer what they believe to be socially desirable rather than their actual experience. However, since the events in the survey took place recently, recall bias should be minimized, while we have no prior indication that social-desirability bias would be of significant concern within our study population. Third, selection bias must be considered due to the sampling method, where those more interested in our study topic might have responded to the invitation. Nonetheless, we believe the key volunteers who helped to disseminate the survey improved the approachability of the survey as it was provided through a trusted source within the community. Fourth, we considered Filipino and Indonesian respondents as a collective body of FDWs in Hong Kong, since our interest was in vaccine uptake of FDWs, and not the difference between the two ethnicities. While differences may still exist, we found no significant difference in the rate of vaccine uptake between the two ethnicities. Fifth, our data was collected at a somewhat early stage of vaccine availability in Hong Kong. After our data collection, further evidence emerged addressing the immunogenicity of the two vaccines available in Hong Kong [44,45]. Such reports may have an effect on vaccine uptake, and specifically, type of vaccine taken. Additionally, the type of vaccines available in Hong Kong may also have an effect on vaccine uptake, though we did not ask those unvaccinated about vaccine-type preference and were not able to control for this in our analysis. However, in Hong Kong both a German developed mRNA vaccine, Comirnaty, and a Chinese developed inactivated virus vaccine, CoronaVac, were both equally accessible options to the population; therefore, concerns related to type and location of the vaccine should be minimal. Finally, our questionnaire was developed for exploratory purposes for this unique population and no internationally standardized vaccine attitudes instrument was available, making our findings not directly comparable with previous studies. Future studies are needed for the FDW populations outside of Hong Kong in countries and regions with different societal and policy structures that could influence vaccine uptake differently. A longitudinal study would also help in addressing the causal relationship between variables found in this study to be associated with vaccine uptake. Qualitative studies may also help to understand in more depth the barriers and mechanisms to vaccine uptake within this community.

## 5. Conclusions

This study, the first to assess vaccine uptake among FDWs, found relatively high vaccination rates associated with encouragement from social networks and the attitude of vaccine uptake contributing to the control of virus spread locally. However, associations with concerns of severe side-effects and long working hours with decreased vaccine uptake remain a concern. As FDWs are a large part of the Hong Kong population who have close contact with their employers’ families and other FDWs in the community, they are an important part of maintaining control of the COVID-19 virus in the city. Health promotion campaigns on vaccine safety and efficacy should be tailored specifically to the FDW population to address concerns of vaccine side effects. The government should use a multi-level approach to target the remaining unvaccinated FDWs in Hong Kong.

## Figures and Tables

**Table 1 ijerph-19-05945-t001:** Characteristics among study sample (*n* = 581).

Factors	Frequency	Vaccinated *	
	N (%)	N (%)	*p*-Value
**Demographics**			
Ethnicity			0.331
Filipino	350 (60%)	280 (80%)	
Indonesian	231 (40%	177 (77%)	
Age			0.036
20–34	181 (31%)	134 (74%)	
35–44	308 (53%)	255 (83%)	
45+	92 (16%)	68 (74%)	
Educational attainment			0.324
Up to Secondary school	288 (50%)	229 (80%)	
Post-secondary/Vocational	167 (29%)	125 (75%)	
University/Postgraduate	126 (22%)	103 (82%)	
Marital status			0.047
Never married	153 (26%)	131 (86%)	
Married	270 (47%)	207 (77%)	
Divorced/widowed/separated	158 (27%)	119 (75%)	
Religious affiliation			0.111
Catholic	305 (53%)	243 (78%)	
Islam	205 (35%)	153 (75%)	
Christian/Others	71 (12%)	61 (86%)	
**Employment conditions**			
Employer’s home size			0.068
≤599 sq. ft.	188 (32%)	140 (75%)	
600–999 sq. ft.	242 (42%)	189 (78%)	
≥1000 sq. ft.	581 (26%)	128 (85%)	
Past month daily working hours			<0.001
13 or less hours	268 (46%)	230 (86%)	
14+ hours	313 (54%)	227 (73%)	
Care for special population (children ≤ 15, adults ≥ 65, disabled/chronically ill)	423 (73%)	329 (78%)	0.397
**Health status**			
Self-rated health (0–100) (mean (SD))	81.6 (19.7)	--	0.049
Chronic health conditions	33 (5.7%)	28 (85%)	0.371
**COVID-19 vaccination**			
Have you ever taken up COVID-19 vaccines? (yes)	457 (79%)	--	
How many doses have you taken? ^†^			
1 dose and I will have 2nd dose	52 (11.4%)	--	
1 dose and I will not have 2nd dose	4 (0.9%)	--	
2 doses	401 (87.7%)	--	
Which type of vaccine did you take? ^†^			
CoronaVac	128 (28.0%)	--	
Comirnaty	326 (71.3%)	--	
Not sure	3 (0.7%)	--	
How severe were your side effects? ^†^			
Not at all/very mild/mild	383 (84%)	--	
Moderate	63 (14%)	--	
Severe/very severe	11 (2%)	--	
How likely are you to take 2 doses of free COVID-19 vaccine in the next 6 months? ^‡^			
Very unlikely/unlikely	11 (9%)	--	
Neutral	65 (52%)	--	
Likely/very likely	48 (39%)	--	
Know someone who had serious side effects after taking up COVID-19 vaccines	103 (18%)	72 (70%)	0.017
Top 3 factors most important in receiving the COVID-19 vaccination ^†^			
For my personal health	197 (43%)	--	
To contribute to the control of COVID-19 in Hong Kong	145 (32%)	--	
To protect those around me	52 (11%)	--	
Top 3 factors most important in not receiving the COVID-19 vaccination? ^‡^			
I am concerned about the serious side effects from the vaccine	57 (46%)	--	
I have not had the chance, but plan to be vaccinated	34 (27%)	--	
I want to wait and see how those who have received the vaccine react first	21 (17%)	--	

χ^2^
*p* values are shown for categorical variables while *t*-test *p* values are shown for continuous variables. ^†^ Among those answering “yes” to ever taking a COVID-19 vaccine (*n* = 457). ^‡^ Among those answering “no” to ever taking a COVID-19 vaccine (*n* = 124). * At least 1 dose of a COVID-19 vaccine.

**Table 2 ijerph-19-05945-t002:** Individual, interpersonal, and socio-structural factors related to COVID-19 uptake (*n* = 581).

	Frequency	Vaccinated *	Chi-Square
	N (%)	N (%)	*p*-Value
**Individual Level Factor:**			
*Positive attitudes: “Taking up COVID-19 vaccination…”*			
Can contribute to the control of COVID-19 in Hong Kong	513 (89%)	441 (86%)	<0.001
Can facilitate resumption of cross-boundary travel	495 (85%)	421 (85%)	<0.001
Is highly effective in protecting those around me against COVID-19	480 (83%)	417 (87%)	<0.001
Is highly effective in protecting you from COVID-19	474 (82%)	413 (87%)	<0.001
*Negative attitudes:*			
I am concerned of having severe side-effects from the COVID-19 vaccines	277 (48%)	202 (73%)	0.001
**Interpersonal factor level:**			
Concerning receiving a COVID-19 vaccine, my employer			<0.001
Encouraged me to receive the vaccine	349 (60%)	306 (88%)	
Did not force, encourage, or discourage me	149 (26%)	85 (57%)	
Forced me to receive the vaccine	58 (10%)	51 (88%)	
Discouraged me to receive the vaccine	25 (4%)	15 (60%)	
My domestic helper friends encouraged me to receive COVID-19 vaccination	398 (69%)	336 (84%)	<0.001
My family members encouraged me to receive COVID-19 vaccination	360 (62%)	318 (88%)	<0.001
**Socio-structural level factor:**			
*Incentives: “I was motivated to get the COVID-19 vaccination to be…”*			
Exempt from the required COVID swab testing of domestic helpers	393 (68%)	336 (86%)	<0.001
Eligible for the 100k HKD lucky draw	198 (34%)	170 (86%)	0.003
*Religious beliefs:*			
COVID-19 vaccination would violate my religious beliefs	26 (5%)	17 (65%)	0.097

* At least 1 dose of a COVID-19 vaccine. Frequencies and vaccination rates are among those who agree with the statement.

**Table 3 ijerph-19-05945-t003:** Past month source and frequency for obtaining information about COVID-19 vaccines (*n* = 581).

	Frequency	Vaccinated *	Chi-Square
	N (%)	N (%)	*p*-Value
**Social media (Facebook, WhatsApp, Instagram, etc.)**			0.044
Almost never/seldom	89 (15%)	62 (70%)	
Sometimes	130 (23%)	100 (77%)	
Always	362 (62%)	295 (82%)	
**Your employers**			0.174
Almost never/seldom	165 (28%)	124 (75%)	
Sometimes	248 (43%)	193 (78%)	
Always	168 (29%)	140 (83%)	
**Friends**			0.117
Almost never/seldom	121 (21%)	90 (74%)	
Sometimes	222 (38%)	170 (77%)	
Always	238 (41%)	197 (83%)	
**Family**			<0.001
Almost never/seldom	171 (29%)	126 (74%)	
Sometimes	208 (36%)	155 (75%)	
Always	202 (35%)	176 (87%)	
**News (newspaper, radio, etc.)**			0.069
Almost never/ seldom	191 (33%)	141 (74%)	
Sometimes	170 (29%)	133 (78%)	
Always	220 (38%)	183 (83%)	
**Overall, information you’ve obtained about COVID-19 has been positive or negative?**			<0.001
Equally positive and negative	286 (49%)	204 (71%)	
Negative	61 (11%)	45 (74%)	
Positive	234 (40%)	208 (89%)	
**Frequency of obtaining COVID-19 vaccine-related information IQR ** ** ^†^ **			0.009
<IQR	166 (29%)	119 (72%)	
IQR	284 (49%)	225 (79%)	
>IQR	131 (23%)	113 (86%)	

^†^ Scale was developed using the 5 source/frequency variables (social media, your employers, friends, family, and news). Cronbach’s Alpha 0.809. Scale was further categorized into inter-quartile range low (bottom 25%), average (middle 50%), and high (top 25%). * At least 1 dose of a COVID-19 vaccine.

**Table 4 ijerph-19-05945-t004:** Associations with vaccine uptake among female foreign domestic workers (*n* = 581).

	Unadjusted OR	*p*-Value	Adjusted OR	*p*-Value
Factors	(95% CI)		(95% CI)	
**Demographics**				
Ethnicity				
Filipino	1.00		**--**	
Indonesian	0.82 (0.55–1.23)	0.331	**--**	**--**
Age				
20–34	1.00		1.00	
35–44	**1.69 (1.08–2.63)**	**0.021**	**2.35 (1.28–4.31)**	**0.006**
45+	0.99 (0.56–1.76)	0.983	0.85 (0.38–1.95)	0.709
Educational attainment				
Up to secondary school	1.00		**--**	
Post-secondary/vocational	1.49 (0.65–3.43)	0.249	**--**	**--**
University/postgraduate	2.24 (0.93–5.42)	0.600	**--**	**--**
Marital status				
Never married	1.00		1.00	
Married	**0.55 (0.32–0.94)**	**0.029**	**0.37 (0.18–0.76)**	**0.007**
Divorced/widowed/separated	**0.51 (0.29–0.91)**	**0.024**	**0.29 (0.13–0.64)**	**0.002**
Religious affiliation				
Catholic	1.00		**--**	
Islam	0.75 (0.49–1.14)	0.181	**--**	**--**
Christian/Others	1.56 (0.75–3.21)	0.231	**--**	**--**
**Employment conditions**				
Employer’s home size				
≤599 sq. ft.	1.00		1.00	
600–999 sq. ft.	1.22 (0.78–1.91)	0.379	**1.92 (1.03–3.59)**	**0.039**
≥1000 sq. ft.	**1.91 (1.10–3.31)**	**0.022**	2.07 (0.96–4.45)	0.064
Past month daily working hours				
13 or less hours	**1.00**		**1.00**	
14+ hours	**0.44 (0.29–0.67)**	**<0.001**	**0.54 (0.30–0.96)**	**0.036**
Care for special population (children, elderly, disabled)	0.82 (0.52–1.30)	0.398	--	--
**Health status**				
Self-rated health (0–100)	**1.01 (1.00–1.02)**	**0.051**	1.01 (1.00–1.02)	0.190
Chronic conditions	1.55 (0.59–4.11)	0.375	--	--
**Individual level factors**				
*Positive attitudes towards COVID-19 vaccines: “Vaccination…”*				
Is highly effective in protecting you from COVID-19	**9.69 (6.06–15.51)**	**<0.001**	1.54 (0.57–4.16)	0.395
Is highly effective in protecting those around me against COVID-19	**10.09 (6.25–16.29)**	**<0.001**	2.18 (0.81–5.89)	0.124
Can facilitate resumption of cross-boundary travel	**7.90 (4.82–12.96)**	**<0.001**	0.99 (0.39–2.51)	0.977
Can contribute to the control of COVID-19 in Hong Kong	**18.62 (10.07–34.43)**	**<0.001**	**6.11 (2.27–16.43)**	**<0.001**
*Negative attitude towards COVID-19 vaccines*				
I am concerned of having severe side-effects from the COVID-19 vaccines	**0.52 (0.35–0.78)**	**0.001**	**0.29 (0.16–0.55)**	**<0.001**
**Interpersonal level factors**				
*Social networks*				
Concerning receiving a COVID-19 vaccine, my employer				
Has been neutral	1.00		1.00	
Forced me to receive the vaccine	**5.49 (2.34–12.89)**	**<0.001**	3.24 (1.00–10.54)	0.051
Encouraged me to receive the vaccine	**5.36 (3.40–8.45)**	**<0.001**	**2.05 (1.06–3.95)**	**0.032**
Discouraged me to receive the vaccine	1.13 (0.48–2.68)	0.782	1.25 (0.39–4.00)	0.709
My family members encouraged me to receive the vaccine	**4.47 (2.93–6.81)**	**<0.001**	**2.27 (1.17–4.38)**	**0.015**
My domestic helper friends encouraged me to receive the vaccine	**2.78 (1.85–4.18)**	**<0.001**	0.66 (0.33–1.33)	0.248
Know someone who had serious side effects after taking up the vaccine	**0.56 (0.35–0.91)**	**0.018**	0.59 (0.31–1.13)	0.113
*COVID-19 vaccine-related news*				
Frequency of obtaining COVID-19 vaccine-related information IQR				
<IQR	1.00		1.00	
IQR	**1.51 (0.97–2.35)**	**0.070**	1.25 (0.66–2.34)	0.495
>IQR	**2.48 (1.36–4.52)**	**0.003**	0.73 (0.33–1.63)	0.445
Overall, information you’ve obtained about COVID-19 vaccination has been…				
Negative/equally positive and negative	1.00		1.00	
Positive	**3.15 (1.97–5.04)**	**<0.001**	1.55 (0.87–2.78)	0.139
**Socio-structural level factors**				
*Incentives: “I was motivated to get the COVID-19 vaccination to be…”*				
Exempt from the required COVID swab testing of domestic helpers	**3.26 (2.17–4.92)**	**<0.001**	1.27 (0.64–2.52)	0.491
Eligible for the 100k HKD lucky draw	**2.03 (1.28–3.22)**	**0.003**	1.18 (0.59–2.38)	0.634
*Religious beliefs*				
COVID-19 vaccination would violate my religious beliefs	**0.49 (0.21–1.14)**	**0.097**	**0.19 (0.06–0.65)**	**0.008**
**Days since first survey completed**	**1.01 (1.00–1.02)**	**0.044**	1.01 (0.99–1.02)	0.317

Numbers in the unadjusted OR column in **bold** indicate the variable was included in the adjusted OR model. Numbers in the adjusted OR column in **bold** indicate the variable remained statistically significant in the final model.

## Data Availability

Not applicable.

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
