# Peer review of "Determinants of COVID-19 Vaccine Uptake among Female Foreign Domestic Workers in Hong Kong: A Cross-Sectional Quantitative Survey"

_ijerph, 2022, doi:10.3390/ijerph19105945_

Round 1

Reviewer 1 Report

  1. In table 1, the demographics are listed. Are there other items that could have been measured and were not? why? what could be the implications of not measuring employment status, type of worker, geography, children at home, COVID infection in family and friends, etc.
  2. In the study measures, what procedures were utilized to assess reliability and validity of survey? Why or why not?
  3. What type of power analysis was conducted to estimate sample size and how?
  4. What are the implications for practice, future research, and policy?
  5. The limitations section should be expanded as there are many limitations to the study design and survey content.
  6. How does this study remain similar to what has been found in other studies and how is this study different from other studies and the findings should be considered. Some studies to consider are-
    https://pubmed.ncbi.nlm.nih.gov/33781902/ 
    https://pubmed.ncbi.nlm.nih.gov/35214687/ 
    https://pubmed.ncbi.nlm.nih.gov/34193545/ 

Author Response

Reviewer 1:

Thank you for your helpful comments and suggestions. We have addressed your comments and concerns in the revision process.

  1. In table 1, the demographics are listed. Are there other items that could have been measured and were not? why? what could be the implications of not measuring employment status, type of worker, geography, children at home, COVID infection in family and friends, etc.

Thank you for your careful consideration of relevant variables used in the study and other potential variables that could have an effect on the outcome of interest. We have rewritten the inclusion criteria sentence to more clearly state our study population. It now states:

“Inclusion criteria were female FDWs from the Philippines or Indonesia, of any age, currently working in Hong Kong as a FDW for at least one year, and had the ability to complete a self-administered questionnaire in English or Bahasa Indonesia.” (lines 127-130)

Regarding the suggested variables of interest:

  • Employment status and type of worker: The employment status of FDW in Hong Kong is homogenous. We have added a sentence in the introduction to clarify employment status and type of worker of foreign domestic workers in Hong Kong:

“All FDWs must enter Hong Kong under the same working visa scheme in which they are required to repatriate within 2 weeks if their full-time contract is terminated [12].” (lines 70-72)

Reference:

  1. Foreign Domestic Helpers | Immigration Department Available online: https://www.immd.gov.hk/eng/services/visas/foreign_domestic_helpers.html (accessed on 8 May 2022).

  • Geography: We controlled for the ethnicity/country of origin of the respondent. However, within Hong Kong we have no reason to believe that their location would have an effect on vaccine uptake as Hong Kong is a largely urban city contained in a relatively small geographic region. Though, to be sure we did check the reported region of residence within Hong Kong and did not find any significant association with vaccine uptake.
  • Children at home: To address vulnerable populations that the respondent may be taking care of, we asked respondents whether they care for any special populations (children under 15, adults 65 and older, and disabled or chronically ill adults). We used this question to consider whether these vulnerable populations they care for may influence their vaccine uptake, this was already included in our model. 
  • COVID-19 infection in friends and family: We agree with the suggestion that COVID-19 infection of family or friends may influence vaccine uptake, but we unfortunately did not ask this question in our survey, and most of the FDWs were away from their family members at home. However, at the time of our data collection, there had just been roughly 12,000 confirmed COVID-19 cases in Hong Kong, 0.16% of the total population, so the chance of many respondents knowing someone with an infection was relatively low. Furthermore, we asked whether respondents had been encouraged by their family and friends to be vaccinated, which we believe would partially account for their friend’s and family’s experience with COVID-19. We do not have any further obtained variables that we believe warrant consideration, though we do appreciate the reviewer’s consideration in the question.

  1. In the study measures, what procedures were utilized to assess reliability and validity of survey? Why or why not?

Thank you for your question. We would like to address two points which your question has helped us consider:

  • Our questionnaire was developed for exploratory purposes for this unique population and no internationally standardized vaccine attitudes instrument was available, making our findings not directly comparable with previous studies.  Nevertheless, we did pilot-test our survey with FDWs to address any questions that may be confusing or unclear before launching the survey for data collection. We have now more clearly stated this in our Methods section, as follows:

“Both languages of the survey were pilot-tested using cognitive debriefing with FDWs prior to mass distribution of survey to ensure understanding and clarity, and changes were made where needed.” (lines 132-134)

  • We do agree that in a survey relying on self-reported data there may be a social-desirability bias as respondents may feel it is more “correct” to report having taken a vaccine even though they have not.

We have added both limitations in our Discussion which now states:

“Second, as our survey relies on self-reported information, our study is subject to recall bias and social-desirability bias, where respondents answer what they believe to be socially desirable rather than their actual experience. However, since the events in the survey took place recently, recall bias should be minimized, while we have no prior indication that social-desirability bias would be of significant concern within our study population.” (lines 385-390)

“Finally, our questionnaire was developed for exploratory purposes for this unique population and no internationally standardized vaccine attitudes instrument was available, making our findings not directly comparable with previous studies.”  (407-410)

  1. What type of power analysis was conducted to estimate sample size and how?

Given a population size of 373,384 female FDWs, with a 5% margin of error, 95% confidence interval, and 50% response distribution, our recommended sample size was 384. We have now added this sample size calculation to our Materials and Methods section:

“Using a population size of 373,384, a margin of error of 5%, a confidence interval of 95%, and a 50% response distribution, the recommended sample size for this study was at least 384 participants.” (lines 134-136)

  1. What are the implications for practice, future research, and policy?

  • Thank you for pointing out the need to more clearly state the study’s implications. We have now added a short paragraph in the Discussion to specifically address the study’s policy implications. For policy implications, we suggest targeted health promotion campaigns tailored to the FDW population to address concerns of vaccine safety.

In summary, a more targeted approach is warranted to improve vaccine uptake in those who may still be hesitant. A health promotion campaign to educate FDWs on vaccine safety and efficacy should be tailored to the FDW community specifically rather than the general Hong Kong population. Additionally, the role that employers play in the rate of vaccine uptake among FDWs should be further considered. Finding approaches that facilitate the positive influence employers can have over FDW vaccine uptake should be promoted.” (lines 367-373)

  • We also added a few sentences at the end of the limitations paragraph to address the direction of future research. In future research we call for the need of assessing vaccine uptake in FDW populations outside of Hong Kong which may have different societal and policy structures, to conduct longitudinal studies which can establish a causal relationship between predictors and vaccine uptake, and qualitative studies which can understand the barriers to vaccine uptake more in-depth.

“Future studies are needed for the FDW populations outside of Hong Kong in countries and regions with different societal and policy structures that could influence vaccine uptake differently. A longitudinal study would also help in addressing the causal relationship between variables found in this study to be associated with vaccine uptake. Qualitative studies may also help to understand in more depth the barriers and mechanisms to vaccine uptake within this community.” (lines 410-415)

  1. The limitations section should be expanded as there are many limitations to the study design and survey content.

We agree that the limitations section could use further explanation and clarification. As mentioned in some responses above, we have now added the potential for social-desirability bias due to the use of self-reported data and the need for validation of the study’s measurements, as follows:

“Second, as our survey relies on self-reported information, our study is subject to recall bias and social-desirability bias, where respondents answer what they believe to be socially desirable rather than their actual experience. However, since the events in the survey took place recently, recall bias should be minimized, while we have no prior indication that social-desirability bias would be of significant concern within our study population.” (lines 385-390)

“Fifth, our data was collected at a somewhat early stage of vaccine availability in Hong Kong. After our data collection, further evidence emerged addressing the immunogenicity of the two vaccines available in Hong Kong [44,45]. Such reports may have an effect on vaccine uptake, and specifically, type of vaccine taken. Additionally, the type of vaccines available in Hong Kong may also have an effect on vaccine uptake, though we did not ask those unvaccinated about vaccine-type preference and were not able to control for this in our analysis. However, in Hong Kong both a German developed mRNA vaccine, Comirnaty, and a Chinese developed inactivated virus vaccine, CoronaVac, were both equally accessible options to the population; therefore, concerns related to type of and location of vaccine should be minimal.” (lines 398-407)

“Finally, our questionnaire was developed for exploratory purposes for this unique population and no internationally standardized vaccine attitudes instrument was available, making our findings not directly comparable with previous studies.”  (407-410)

References:

  1. Mok, C.K.P.; Cohen, C.A.; Cheng, S.M.S.; Chen, C.; Kwok, K.O.; Yiu, K.; Chan, T.O.; Bull, M.; Ling, K.C.; Dai, Z.; et al. Comparison of the Immunogenicity of BNT162b2 and CoronaVac COVID-19 Vaccines in Hong Kong. Respirology 2022, 27, 301–310, doi:10.1111/RESP.14191.
  2. Cheng, S.M.S.; Mok, C.K.P.; Leung, Y.W.Y.; Ng, S.S.; Chan, K.C.K.; Ko, F.W.; Chen, C.; Yiu, K.; Lam, B.H.S.; Lau, E.H.Y.; et al. Neutralizing Antibodies against the SARS-CoV-2 Omicron Variant BA.1 Following Homologous and Heterologous CoronaVac or BNT162b2 Vaccination. Nat. Med. 2022 283 2022, 28, 486–489, doi:10.1038/s41591-022-01704-7.

  1. How does this study remain similar to what has been found in other studies and how is this study different from other studies and the findings should be considered. Some studies to consider are-
    https://pubmed.ncbi.nlm.nih.gov/33781902/ 
    https://pubmed.ncbi.nlm.nih.gov/35214687/ 
    https://pubmed.ncbi.nlm.nih.gov/34193545/

Thank you for your helpful questions. We have expanded both the Introduction and Discussion sections to introduce related studies and compare our findings with similar studies. In the Introduction, we added a paragraph to introduce previous vaccine-related literature in Hong Kong specifically. We also expanded on the final paragraph which introduced the socio-ecological model to describe the findings from previous studies in more detail.

Introduction:

Vaccine hesitancy and vaccine uptake have been previously studied in the general Hong Kong population. A serial cross-sectional survey of Cantonese-speaking working adults found that a decrease in willingness to be vaccinated was associated with an increase in concern for vaccine safety [8]. Another serial cross-sectional study of Cantonese-speaking adults aged 18 and above found prior to the “vaccination for all” campaign, vaccine hesitancy was highest in young adults (aged 18-34), while after the launch of the campaign hesitancy was highest in older adults (aged ≥ 65) [9]. Furthermore, higher vaccine uptake was positively associated with middle-age (aged 35-59), higher educational attainment, higher confidence in COVID-19 vaccines, and greater trust in their own ability to prevent infection [9]. However, both studies only included Cantonese-speaking permanent residents of Hong Kong, excluding ethnic minority populations and those without permanent residency status. A single study on determinants of COVID-19 vaccine uptake in the South Asian ethnic minority adult population of Hong Kong was conducted in May 2021 and found a relatively low vaccine uptake rate of 33.1% among its 245 respondents [10]. Those with positive attitudes toward vaccination, perceived support from significant others, and perceived higher behavioral control to receive vaccination were positively associated with vaccine uptake, while higher exposure to information about deaths and serious conditions caused by COVID-19 vaccination was associated with decreased vaccine uptake [10].” (lines 48-66)

“The socio-ecological model has been used to assess COVID-19 vaccine acceptance and uptake in several studies [10,22–25]. The socio-ecological model is a theoretical framework commonly used in social sciences and public health which considers both individual and societal factors contributing to health outcomes [26]. The socio-ecological model is comprised of sociodemographic factors, individual, interpersonal, and socio-structural factors [26]. Informed by the socio-ecological model, a scoping review including 50 studies assessed factors which influenced public attitudes towards COVID-19 vaccination globally and found contributing factors at each level [22]. At the sociodemographic level, factors associated with positive attitudes towards vaccination included female sex, age older than 25, higher educational attainment, high income status, chronic diseases, being employed, and being married with children [22]. At the individual level, those who trusted the government, healthcare system, and companies producing the vaccines had more positive attitudes towards vaccination, while those with conspiracy theory beliefs and those with religious reasons for refusing the vaccine were more vaccine hesitant [22]. At the interpersonal level, positive messaging from healthcare professionals, family, and friends were associated with positive attitudes towards vaccination, while one study found employer’s recommendation may also contribute to a positive vaccine attitude [22,27]. Finally, at the socio-structural level, both traditional news media and social media were contributing factors regarding vaccination attitudes [22]. Addressing determinants at multiple levels may increase the likelihood of changing health behaviors. To the best of our knowledge, there are no studies examining the determinants of vaccine uptake among FDWs anywhere in the world. As there are over 8.5 million female FDWs globally [28], the findings of this study may help improve vaccine uptake for FDWs in Hong Kong and in other countries.” (lines 95-118)

Discussion:

“A study in six European countries found varying effectiveness of mandatory COVID-19 certification (vaccination, negative test, proof of recovery) on vaccine uptake, with those younger than 30 having the highest increase in vaccine uptake post certification implementation [31]. This differs from our sample which found the highest vaccine uptake among 35-44 year-old, even though this could be due to different societal factors.” (lines 281-285)

“The risks and issues with financial incentives have been previously discussed, with researchers recommending incentives like access to certain facilities to only vaccinated people as more sustainable, than cash incentives, in the long term [35].” (lines 295-297)

“Among the demographic characteristics, never married FDWs were more likely to take up vaccination, which differed from the South Asian ethnic minority group in Hong Kong, the general Hong Kong population, and the general findings of a scoping review where being married or cohabitating was positively associated with vaccine uptake/acceptance [8,10,22].” (lines 306-310)

“A multi-country survey early in the pandemic found that 48% of respondents would accept vaccination if their employer recommended it, though rates varied by country, with China reporting the highest acceptance and Russia the lowest [27].” (337-339)

One study from the Czech Republic found higher rates of vaccine hesitancy in those identifying as spiritual but with non-religious affiliation than those with religious affiliation [39]. More in-depth studies are needed to understand how exactly religious beliefs and religiosity may affect vaccine uptake and how to most effectively address these concerns.” (lines 346-350)

References:

  1. Wang, K.; Wong, E.L.Y.; Ho, K.F.; Cheung, A.W.L.; Yau, P.S.Y.; Dong, D.; Wong, S.Y.S.; Yeoh, E.K. Change of Willingness to Accept COVID-19 Vaccine and Reasons of Vaccine Hesitancy of Working People at Different Waves of Local Epidemic in Hong Kong, China: Repeated Cross-Sectional Surveys. Vaccines 2021, Vol. 9, Page 62 2021, 9, 62, doi:10.3390/VACCINES9010062.
  2. Xiao, J.; Cheung, J.K.; Wu, P.; Ni, M.Y.; Cowling, B.J.; Liao, Q. Temporal Changes in Factors Associated with COVID-19 Vaccine Hesitancy and Uptake among Adults in Hong Kong: Serial Cross-Sectional Surveys. Lancet Reg. Heal. - West. Pacific 2022, 23, 100441, doi:10.1016/J.LANWPC.2022.100441/ATTACHMENT/0958D23D-7932-439B-A4BC-373A87C61BAD/MMC6.PDF.
  3. Singh, A.; Lai, A.H.Y.; Wang, J.; Asim, S.; Shing-Fong Chan, P.; Wang, Z.; Yeoh, E.K. Multilevel Determinants of COVID-19 Vaccine Uptake Among South Asian Ethnic Minorities in Hong Kong: Cross-Sectional Web-Based Survey. JMIR Public Heal. Surveill 2021;7(11)e31707 https//publichealth.jmir.org/2021/11/e31707 2021, 7, e31707, doi:10.2196/31707.
  4. Al-Jayyousi, G.F.; Sherbash, M.A.M.; Ali, L.A.M.; El-Heneidy, A.; Alhussaini, N.W.Z.; Elhassan, M.E.A.; Nazzal, M.A. Factors Influencing Public Attitudes towards COVID-19 Vaccination: A Scoping Review Informed by the Socio-Ecological Model. Vaccines 2021, Vol. 9, Page 548 2021, 9, 548, doi:10.3390/VACCINES9060548.
  5. Latkin, C.; Dayton, L.A.; Yi, G.; Konstantopoulos, A.; Park, J.; Maulsby, C.; Kong, X. COVID-19 Vaccine Intentions in the United States, a Social-Ecological Framework. Vaccine 2021, 39, 2288, doi:10.1016/J.VACCINE.2021.02.058.
  6. Zhang, K.C.; Fang, Y.; Cao, H.; Chen, H.; Hu, T.; Chen, Y.; Zhou, X.; Wang, Z. Behavioral Intention to Receive a COVID-19 Vaccination Among Chinese Factory Workers: Cross-Sectional Online Survey. J. Med. Internet Res. 2021, 23, doi:10.2196/24673.
  7. Huang, X.; Yu, M.; Fu, G.; Lan, G.; Li, L.; Yang, J.; Qiao, Y.; Zhao, J.; Qian, H.-Z.; Zhang, X.; et al. Willingness to Receive COVID-19 Vaccination Among People Living With HIV and AIDS in China: Nationwide Cross-Sectional Online Survey. JMIR public Heal. Surveill. 2021, 7, e31125, doi:10.2196/31125.
  8. KR, M.; D, B.; A, S.; K, G. An Ecological Perspective on Health Promotion Programs. Health Educ. Q. 1988, 15, 351–377, doi:10.1177/109019818801500401.
  9. Lazarus, J. V.; Ratzan, S.C.; Palayew, A.; Gostin, L.O.; Larson, H.J.; Rabin, K.; Kimball, S.; El-Mohandes, A. A Global Survey of Potential Acceptance of a COVID-19 Vaccine. Nat. Med. 2021, 27, 225, doi:10.1038/S41591-020-1124-9.
  10. Gallotti, M. Migrant Domestic Workers Across the World: Global and Regional Estimates; Geneva, 2016;
  11. Mills, M.C.; Rüttenauer, T. The Effect of Mandatory COVID-19 Certificates on Vaccine Uptake: Synthetic-Control Modelling of Six Countries. Lancet Public Heal. 2022, 7, e15–e22, doi:10.1016/S2468-2667(21)00273-5/ATTACHMENT/D55AD999-A00D-4AB6-9023-7898B97DF3AF/MMC1.PDF.
  12. Volpp, K.G.; Cannuscio, C.C. Incentives for Immunity — Strategies for Increasing Covid-19 Vaccine Uptake. N. Engl. J. Med. 2021, 385, e1, doi:10.1056/NEJMP2107719/SUPPL_FILE/NEJMP2107719_DISCLOSURES.PDF.
  13. Kosarkova, A.; Malinakova, K.; van Dijk, J.P.; Tavel, P. Vaccine Refusal in the Czech Republic Is Associated with Being Spiritual but Not Religiously Affiliated. Vaccines 2021, Vol. 9, Page 1157 2021, 9, 1157, doi:10.3390/VACCINES9101157.

Reviewer 2 Report

The Introduction explains the socio-ecological model as the paradigm used by the authors to explain vaccine uptake among FDW's in Hong Kong. I think the model is very adequate.

Still, while explaining vaccine acceptance, the authors cite a rather general article (citation 22).

I think that should be completed with citing actual studies that used socio-demographic factors, and also psychosocial factors, to explain vaccine acceptance/vaccine hesitancy.

Below is a list of articles that present mechanisms that can be invoked as theoretical prerequisites for explaining hesitancy, respectively the acceptance of the vaccine:

  1. Studies that invoke socio-demographic factors: Vaccines | Free Full-Text | A Scoping Review to Find Out Worldwide COVID-19 Vaccine Hesitancy and Its Underlying Determinants (mdpi.com) ; Vaccines | Free Full-Text | Vaccine Refusal in the Czech Republic Is Associated with Being Spiritual but Not Religiously Affiliated (mdpi.com)
  2. Studies regarding psychosocial factors related to pro-vaccination attitudes: the study in the following link (Vaccines | Free Full-Text | Vaccinating against COVID-19: The Correlation between Pro-Vaccination Attitudes and the Belief That Our Peers Want to Get Vaccinated (mdpi.com)) demonstrates a correlation between pro-vaccination attitudes and the belief that others like us want to get vaccinated too. The next one takes into account religiosity: The role of religiosity in COVID-19 vaccine hesitancy - PubMed (nih.gov)
  3. Trust in vaccine and attitudes towards conspiracy theories as explanations for vaccine hesitancy/acceptance: Vaccines | Free Full-Text | The Mediating Roles of Medical Mistrust, Knowledge, Confidence and Complacency of Vaccines in the Pathways from Conspiracy Beliefs to Vaccine Hesitancy (mdpi.com) ; Vaccines | Free Full-Text | Attitudes and Intentions toward COVID-19 Vaccination among Health Professions Students and Faculty in Qatar (mdpi.com)

Author Response

Reviewer 2:

Thank you for your helpful comments and suggestions. We have addressed your comments and concerns in the revision process.

The Introduction explains the socio-ecological model as the paradigm used by the authors to explain vaccine uptake among FDW's in Hong Kong. I think the model is very adequate.

Still, while explaining vaccine acceptance, the authors cite a rather general article (citation 22).

I think that should be completed with citing actual studies that used socio-demographic factors, and also psychosocial factors, to explain vaccine acceptance/vaccine hesitancy.

Below is a list of articles that present mechanisms that can be invoked as theoretical prerequisites for explaining hesitancy, respectively the acceptance of the vaccine:

  1. Studies that invoke socio-demographic factors: Vaccines | Free Full-Text | A Scoping Review to Find Out Worldwide COVID-19 Vaccine Hesitancy and Its Underlying Determinants (mdpi.com) ; Vaccines | Free Full-Text | Vaccine Refusal in the Czech Republic Is Associated with Being Spiritual but Not Religiously Affiliated (mdpi.com)
  2. Studies regarding psychosocial factors related to pro-vaccination attitudes: the study in the following link (Vaccines | Free Full-Text | Vaccinating against COVID-19: The Correlation between Pro-Vaccination Attitudes and the Belief That Our Peers Want to Get Vaccinated (mdpi.com)) demonstrates a correlation between pro-vaccination attitudes and the belief that others like us want to get vaccinated too. The next one takes into account religiosity: The role of religiosity in COVID-19 vaccine hesitancy - PubMed (nih.gov)
  3. Trust in vaccine and attitudes towards conspiracy theories as explanations for vaccine hesitancy/acceptance: Vaccines | Free Full-Text | The Mediating Roles of Medical Mistrust, Knowledge, Confidence and Complacency of Vaccines in the Pathways from Conspiracy Beliefs to Vaccine Hesitancy (mdpi.com); Vaccines | Free Full-Text | Attitudes and Intentions toward COVID-19 Vaccination among Health Professions Students and Faculty in Qatar (mdpi.com)

We appreciate the list of helpful studies that the reviewer provided us with and agree with the insightful suggestion that this part of our introduction warrants further explanation of the actual findings when considering the socio-ecological model. In an effort to be more succinct while adding important information to this paragraph, we largely used the evidence from a scoping review with a global perspective to address factors influencing vaccine hesitancy and acceptance at different levels of the socio-ecological model. However, we did include the suggested study on vaccine refusal in the Czech Republic being associated with being spiritual but not religiously affiliated in the Discussion of our manuscript. This paragraph in the Introduction now states:

“The socio-ecological model has been used to assess COVID-19 vaccine acceptance and uptake in several studies [10,22–25]. The socio-ecological model is a theoretical framework commonly used in social sciences and public health which considers both individual and societal factors contributing to health outcomes [26]. The socio-ecological model is comprised of sociodemographic factors, individual, interpersonal, and socio-structural factors [26]. Informed by the socio-ecological model, a scoping review including 50 studies assessed factors which influenced public attitudes towards COVID-19 vaccination globally and found contributing factors at each level [22]. At the sociodemographic level, factors associated with positive attitudes towards vaccination included female sex, age older than 25, higher educational attainment, high income status, chronic diseases, being employed, and being married with children [22]. At the individual level, those who trusted the government, healthcare system, and companies producing the vaccines had more positive attitudes towards vaccination, while those with conspiracy theory beliefs and those with religious reasons for refusing the vaccine were more vaccine hesitant [22]. At the interpersonal level, positive messaging from healthcare professionals, family, and friends were associated with positive attitudes towards vaccination, while one study found employer’s recommendation may also contribute to a positive vaccine attitude [22,27]. Finally, at the socio-structural level, both traditional news media and social media were contributing factors regarding vaccination attitudes [22]. Addressing determinants at multiple levels may increase the likelihood of changing health behaviors. To the best of our knowledge, there are no studies examining the determinants of vaccine uptake among FDWs anywhere in the world. As there are over 8.5 million female FDWs globally [28], the findings of this study may help improve vaccine uptake for FDWs in Hong Kong and in other countries.” (lines 95-118)

References:

  1. Singh, A.; Lai, A.H.Y.; Wang, J.; Asim, S.; Shing-Fong Chan, P.; Wang, Z.; Yeoh, E.K. Multilevel Determinants of COVID-19 Vaccine Uptake Among South Asian Ethnic Minorities in Hong Kong: Cross-Sectional Web-Based Survey. JMIR Public Heal. Surveill 2021;7(11)e31707 https//publichealth.jmir.org/2021/11/e31707 2021, 7, e31707, doi:10.2196/31707.
  2. Wong, W. No Mandatory Jab for Helpers, but Another Test Needed Available online: https://news.rthk.hk/rthk/en/component/k2/1590257-20210511.htm (accessed on 3 November 2021).
  3. Al-Jayyousi, G.F.; Sherbash, M.A.M.; Ali, L.A.M.; El-Heneidy, A.; Alhussaini, N.W.Z.; Elhassan, M.E.A.; Nazzal, M.A. Factors Influencing Public Attitudes towards COVID-19 Vaccination: A Scoping Review Informed by the Socio-Ecological Model. Vaccines 2021, Vol. 9, Page 548 2021, 9, 548, doi:10.3390/VACCINES9060548.
  4. Latkin, C.; Dayton, L.A.; Yi, G.; Konstantopoulos, A.; Park, J.; Maulsby, C.; Kong, X. COVID-19 Vaccine Intentions in the United States, a Social-Ecological Framework. Vaccine 2021, 39, 2288, doi:10.1016/J.VACCINE.2021.02.058.
  5. Zhang, K.C.; Fang, Y.; Cao, H.; Chen, H.; Hu, T.; Chen, Y.; Zhou, X.; Wang, Z. Behavioral Intention to Receive a COVID-19 Vaccination Among Chinese Factory Workers: Cross-Sectional Online Survey. J. Med. Internet Res. 2021, 23, doi:10.2196/24673.
  6. Huang, X.; Yu, M.; Fu, G.; Lan, G.; Li, L.; Yang, J.; Qiao, Y.; Zhao, J.; Qian, H.-Z.; Zhang, X.; et al. Willingness to Receive COVID-19 Vaccination Among People Living With HIV and AIDS in China: Nationwide Cross-Sectional Online Survey. JMIR public Heal. Surveill. 2021, 7, e31125, doi:10.2196/31125.
  7. KR, M.; D, B.; A, S.; K, G. An Ecological Perspective on Health Promotion Programs. Health Educ. Q. 1988, 15, 351–377, doi:10.1177/109019818801500401.
  8. Lazarus, J. V.; Ratzan, S.C.; Palayew, A.; Gostin, L.O.; Larson, H.J.; Rabin, K.; Kimball, S.; El-Mohandes, A. A Global Survey of Potential Acceptance of a COVID-19 Vaccine. Nat. Med. 2021, 27, 225, doi:10.1038/S41591-020-1124-9.
  9. Gallotti, M. Migrant Domestic Workers Across the World: Global and Regional Estimates; Geneva, 2016;

Reviewer 3 Report

it would be interesting to continue the field with a study design that allow for causal inference

What is the main question addressed by the research?

The study seeks to understand determinants of vaccine uptake among female foreign domestic workers in Hong Kong.

Is it relevant and interesting? It’s interesting but no relevant for the type of the study (Cross-sectional). This type of study should be reserved as an initial approach in case of unexpected results to be further investigated with a verification made with more accurate studies.

How original is the topic? The study of determinants of vaccine uptake has low originality* .  

What does it add to the subject area compared with other published material? It adds knowledge about the determinants in a specific study group.

Is the paper well written? Yes

Is the text clear and easy to read? Yes

Are the conclusions consistent with the evidence and arguments presented? Yes, but the results intuitive and the scientific evidence is low.

Do they address the main question posed? Yes

*

- Singh A, Lai AHY, Wang J, Asim S, Chan PS, Wang Z, Yeoh EK. Multilevel Determinants of COVID-19 Vaccine Uptake Among South Asian Ethnic Minorities in Hong Kong: Cross-sectional Web-Based Survey. JMIR Public Health Surveill. 2021 Nov 9;7(11):e31707.

- Luk TT, Zhao S, Wu Y, Wong JY, Wang MP, Lam TH. Prevalence and determinants of SARS-CoV-2 vaccine hesitancy in Hong Kong: A population-based survey. Vaccine. 2021 Jun 16;39(27):3602-3607.

- Zhang KC, Fang Y, Cao H, Chen H, Hu T, Chen Y, Zhou X, Wang Z. Behavioral Intention to Receive a COVID-19 Vaccination Among Chinese Factory Workers: Cross-sectional Online Survey. J Med Internet Res. 2021 Mar 9;23(3):e24673. doi: 10.2196/24673.

- Alemayehu A, Yusuf M, Demissie A, Abdullahi Y. Determinants of COVID-19 vaccine uptake and barriers to being vaccinated among first-round eligibles for COVID-19 vaccination in Eastern Ethiopia: A community based cross-sectional study. SAGE Open Med. 2022 Feb 8;10:20503121221077585. doi: 10.1177/20503121221077585.

- Rane MS, Kochhar S, Poehlein E, You W, Robertson MM, Zimba R, Westmoreland DA, Romo ML, Kulkarni SG, Chang M, Berry A, Parcesepe AM, Maroko AR, Grov C, Nash D; CHASING COVID Cohort Study Team. Determinants and Trends of COVID-19 Vaccine Hesitancy and Vaccine Uptake in a National Cohort of US Adults: A Longitudinal Study. Am J Epidemiol. 2022 Mar 24;191(4):570-583. doi: 10.1093/aje/kwab293.

- Gray A, Fisher CB. Determinants of COVID-19 Vaccine Uptake in Adolescents 12-17 Years Old: Examining Pediatric Vaccine Hesitancy Among Racially Diverse Parents in the United States. Front Public Health. 2022 Mar 22;10:844310. doi: 10.3389/fpubh.2022.844310.

Author Response

Reviewer 3:

Thank you for your helpful comments and suggestions. We have addressed your comments and concerns in the revision process.

it would be interesting to continue the field with a study design that allow for causal inference

We totally agree that a follow-up study would be very interesting and informative for this population. We will take into consideration the feasibility of additionally conducting a follow-up survey with our study population for a future study. We have also included this suggestion in the Discussion on the direction of future research.

“Future studies are needed for the FDW populations outside of Hong Kong in countries and regions with different societal and policy structures that could influence vaccine uptake differently. A longitudinal study would also help in addressing the causal relationship between variables found in this study to be associated with vaccine uptake. Qualitative studies may also help to understand in more depth the barriers and mechanisms to vaccine uptake within this community.” (lines 410-415)

What is the main question addressed by the research?

The study seeks to understand determinants of vaccine uptake among female foreign domestic workers in Hong Kong.

Is it relevant and interesting? It’s interesting but no relevant for the type of the study (Cross-sectional). This type of study should be reserved as an initial approach in case of unexpected results to be further investigated with a verification made with more accurate studies.

What does it add to the subject area compared with other published material? It adds knowledge about the determinants in a specific study group.

Is the paper well written? Yes

Is the text clear and easy to read? Yes

Are the conclusions consistent with the evidence and arguments presented? Yes, but the results intuitive and the scientific evidence is low.

Thank you for your comment. We have revised the Discussion to further consider how our findings relate to other research studies, as below:

“A study in six European countries found varying effectiveness of mandatory COVID-19 certification (vaccination, negative test, proof of recovery) on vaccine uptake, with those younger than 30 having the highest increase in vaccine uptake post certification implementation [31]. This differs from our sample which found the highest vaccine uptake among 35-44 year-old, even though this could be due to different societal factors.” (lines 281-285)

“The risks and issues with financial incentives have been previously discussed, with researchers recommending incentives like access to certain facilities to only vaccinated people as more sustainable, than cash incentives, in the long term [35].” (lines 295-297)

“Among the demographic characteristics, never married FDWs were more likely to take up vaccination, which differed from the South Asian ethnic minority group in Hong Kong, the general Hong Kong population, and the general findings of a scoping review where being married or cohabitating was positively associated with vaccine uptake/acceptance [8,10,22].” (lines 306-310)

“A multi-country survey early in the pandemic found that 48% of respondents would accept vaccination if their employer recommended it, though rates varied by country, with China reporting the highest acceptance and Russia the lowest [27].” (337-339)

One study from the Czech Republic found higher rates of vaccine hesitancy in those identifying as spiritual but with non-religious affiliation than those with religious affiliation [39]. More in-depth studies are needed to understand how exactly religious beliefs and religiosity may affect vaccine uptake and how to most effectively address these concerns.” (lines 346-350)

References:

  1. Wang, K.; Wong, E.L.Y.; Ho, K.F.; Cheung, A.W.L.; Yau, P.S.Y.; Dong, D.; Wong, S.Y.S.; Yeoh, E.K. Change of Willingness to Accept COVID-19 Vaccine and Reasons of Vaccine Hesitancy of Working People at Different Waves of Local Epidemic in Hong Kong, China: Repeated Cross-Sectional Surveys. Vaccines 2021, Vol. 9, Page 62 2021, 9, 62, doi:10.3390/VACCINES9010062.
  2. Xiao, J.; Cheung, J.K.; Wu, P.; Ni, M.Y.; Cowling, B.J.; Liao, Q. Temporal Changes in Factors Associated with COVID-19 Vaccine Hesitancy and Uptake among Adults in Hong Kong: Serial Cross-Sectional Surveys. Lancet Reg. Heal. - West. Pacific 2022, 23, 100441, doi:10.1016/J.LANWPC.2022.100441/ATTACHMENT/0958D23D-7932-439B-A4BC-373A87C61BAD/MMC6.PDF.
  3. Singh, A.; Lai, A.H.Y.; Wang, J.; Asim, S.; Shing-Fong Chan, P.; Wang, Z.; Yeoh, E.K. Multilevel Determinants of COVID-19 Vaccine Uptake Among South Asian Ethnic Minorities in Hong Kong: Cross-Sectional Web-Based Survey. JMIR Public Heal. Surveill 2021;7(11)e31707 https//publichealth.jmir.org/2021/11/e31707 2021, 7, e31707, doi:10.2196/31707.
  4. Al-Jayyousi, G.F.; Sherbash, M.A.M.; Ali, L.A.M.; El-Heneidy, A.; Alhussaini, N.W.Z.; Elhassan, M.E.A.; Nazzal, M.A. Factors Influencing Public Attitudes towards COVID-19 Vaccination: A Scoping Review Informed by the Socio-Ecological Model. Vaccines 2021, Vol. 9, Page 548 2021, 9, 548, doi:10.3390/VACCINES9060548.
  5. Latkin, C.; Dayton, L.A.; Yi, G.; Konstantopoulos, A.; Park, J.; Maulsby, C.; Kong, X. COVID-19 Vaccine Intentions in the United States, a Social-Ecological Framework. Vaccine 2021, 39, 2288, doi:10.1016/J.VACCINE.2021.02.058.
  6. Zhang, K.C.; Fang, Y.; Cao, H.; Chen, H.; Hu, T.; Chen, Y.; Zhou, X.; Wang, Z. Behavioral Intention to Receive a COVID-19 Vaccination Among Chinese Factory Workers: Cross-Sectional Online Survey. J. Med. Internet Res. 2021, 23, doi:10.2196/24673.
  7. Huang, X.; Yu, M.; Fu, G.; Lan, G.; Li, L.; Yang, J.; Qiao, Y.; Zhao, J.; Qian, H.-Z.; Zhang, X.; et al. Willingness to Receive COVID-19 Vaccination Among People Living With HIV and AIDS in China: Nationwide Cross-Sectional Online Survey. JMIR public Heal. Surveill. 2021, 7, e31125, doi:10.2196/31125.
  8. KR, M.; D, B.; A, S.; K, G. An Ecological Perspective on Health Promotion Programs. Health Educ. Q. 1988, 15, 351–377, doi:10.1177/109019818801500401.
  9. Lazarus, J. V.; Ratzan, S.C.; Palayew, A.; Gostin, L.O.; Larson, H.J.; Rabin, K.; Kimball, S.; El-Mohandes, A. A Global Survey of Potential Acceptance of a COVID-19 Vaccine. Nat. Med. 2021, 27, 225, doi:10.1038/S41591-020-1124-9.
  10. Gallotti, M. Migrant Domestic Workers Across the World: Global and Regional Estimates; Geneva, 2016;
  11. Mills, M.C.; Rüttenauer, T. The Effect of Mandatory COVID-19 Certificates on Vaccine Uptake: Synthetic-Control Modelling of Six Countries. Lancet Public Heal. 2022, 7, e15–e22, doi:10.1016/S2468-2667(21)00273-5/ATTACHMENT/D55AD999-A00D-4AB6-9023-7898B97DF3AF/MMC1.PDF.
  12. Volpp, K.G.; Cannuscio, C.C. Incentives for Immunity — Strategies for Increasing Covid-19 Vaccine Uptake. N. Engl. J. Med. 2021, 385, e1, doi:10.1056/NEJMP2107719/SUPPL_FILE/NEJMP2107719_DISCLOSURES.PDF.
  13. Kosarkova, A.; Malinakova, K.; van Dijk, J.P.; Tavel, P. Vaccine Refusal in the Czech Republic Is Associated with Being Spiritual but Not Religiously Affiliated. Vaccines 2021, Vol. 9, Page 1157 2021, 9, 1157, doi:10.3390/VACCINES9101157.

Do they address the main question posed? Yes

How original is the topic? The study of determinants of vaccine uptake has low originality* .  

*

- Singh A, Lai AHY, Wang J, Asim S, Chan PS, Wang Z, Yeoh EK. Multilevel Determinants of COVID-19 Vaccine Uptake Among South Asian Ethnic Minorities in Hong Kong: Cross-sectional Web-Based Survey. JMIR Public Health Surveill. 2021 Nov 9;7(11):e31707.

- Luk TT, Zhao S, Wu Y, Wong JY, Wang MP, Lam TH. Prevalence and determinants of SARS-CoV-2 vaccine hesitancy in Hong Kong: A population-based survey. Vaccine. 2021 Jun 16;39(27):3602-3607.

- Zhang KC, Fang Y, Cao H, Chen H, Hu T, Chen Y, Zhou X, Wang Z. Behavioral Intention to Receive a COVID-19 Vaccination Among Chinese Factory Workers: Cross-sectional Online Survey. J Med Internet Res. 2021 Mar 9;23(3):e24673. doi: 10.2196/24673.

- Alemayehu A, Yusuf M, Demissie A, Abdullahi Y. Determinants of COVID-19 vaccine uptake and barriers to being vaccinated among first-round eligibles for COVID-19 vaccination in Eastern Ethiopia: A community based cross-sectional study. SAGE Open Med. 2022 Feb 8;10:20503121221077585. doi: 10.1177/20503121221077585.

- Rane MS, Kochhar S, Poehlein E, You W, Robertson MM, Zimba R, Westmoreland DA, Romo ML, Kulkarni SG, Chang M, Berry A, Parcesepe AM, Maroko AR, Grov C, Nash D; CHASING COVID Cohort Study Team. Determinants and Trends of COVID-19 Vaccine Hesitancy and Vaccine Uptake in a National Cohort of US Adults: A Longitudinal Study. Am J Epidemiol. 2022 Mar 24;191(4):570-583. doi: 10.1093/aje/kwab293.

- Gray A, Fisher CB. Determinants of COVID-19 Vaccine Uptake in Adolescents 12-17 Years Old: Examining Pediatric Vaccine Hesitancy Among Racially Diverse Parents in the United States. Front Public Health. 2022 Mar 22;10:844310. doi: 10.3389/fpubh.2022.844310.

We thank the reviewer for their comment and list of studies which relate to the topic of our study. We agree that the topic of vaccine uptake in general has been done many times, especially in the COVID-19 era. However, given the uniqueness of our study population, we believe a relevant and important research gap remains. In the manuscript, to make this point we have a paragraph in the Introduction (lines 67-94) which introduces the unique position and vulnerabilities of the FDW population in Hong Kong. This paragraph points out their inability to seek permanent residency status in Hong Kong, unlike other ethnic minority populations (lines 73-75), and the vulnerability they are subjected to being required to live in the same home as their employer (lines 75-77). We also wanted to point out the difficulties they have faced during the COVID-19 pandemic which attribute to their population being of particular interest (lines 77-94).

Reviewer 4 Report

The manuscript presents a study on factors influencing the decision to vaccinate with at least one dose of a COVID-19 vaccine. The study was conducted among female foreign domestic workers, the largest group of ethnic minorities in Hong Kong. It was the first to assess vaccine uptake among this group of people. The research was carried out transparently and comprehensively. The authors presented a critical discussion of the obtained research results. The results and conclusions are essential in the context of actions that the government may take to increase vaccination in this group of people, not only in Hong Kong but anywhere in the world.

Moreover, addressing determinants at multiple levels may increase the likelihood of changing health behaviors. This is an interesting study in which a comprehensive dataset has been collected. The manuscript is clear, concise, and presented in a well-structured manner. The introduction is relevant, and sufficient information about the previous study findings is presented for readers to follow the present study rationale and procedures. Overall, the results are clear and compelling. The cited references are current, mainly within the last 1-2 years.

Several changes are required before this manuscript can be considered for publication in the International Journal of Environmental Research and Public Health. Please find below my comments which I believe would help improve the quality of the manuscript.

  1. Names of the COVID-19 vaccine that people in the study declared to have been vaccinated are inconsistent throughout the manuscript. The Materials and Methods section lists the names of companies producing vaccines against COVID-19: Sinovac-Biotech and BioNTech-Fosun Pharma, while the Results section contains the name BioN-Tech vaccine. The names should be uniform throughout the text of the manuscript. It is appropriate to mention the name of the preparation, not just the name of the company producing the preparation. Moreover, for BNT162b2 (BioNTech-Fosun Pharma) vaccine, it is worth mentioning that it is an mRNA vaccine, and CoronaVac (Sinovac-Biotech) vaccine, that it is an inactivated pathogen vaccine.
  2. Have studies conducted after vaccination with a different type of vaccine, e.g., mRNA-, adenovirus-, protein-based or whole virus vaccine, could alter the results presented in the manuscript. Do the authors suppose that the perception of the safety of using non-mRNA-based or inactivated pathogen vaccines is similar?
  3. Apart from those listed in the manuscript's discussion section, what are the study's limitations? 

Minor comments

  1. Notation of numbers are misspelled throughout the text of the manuscript. Decimals should be written: 5.7 instead of 5∙7 (Line 164).
  2. Please reformat the Tables to improve their readability. There should be a space between the footnotes Table1 and Table2 and the description of the tables.
  3. Grammatical errors and typos should be corrected throughout the manuscript.

Given these shortcomings, the manuscript requires minor revisions.

Author Response

Reviewer 4:

Thank you for your encouraging words and helpful comments and suggestions. We have addressed your comments and concerns in the revision process.

The manuscript presents a study on factors influencing the decision to vaccinate with at least one dose of a COVID-19 vaccine. The study was conducted among female foreign domestic workers, the largest group of ethnic minorities in Hong Kong. It was the first to assess vaccine uptake among this group of people. The research was carried out transparently and comprehensively. The authors presented a critical discussion of the obtained research results. The results and conclusions are essential in the context of actions that the government may take to increase vaccination in this group of people, not only in Hong Kong but anywhere in the world.

Moreover, addressing determinants at multiple levels may increase the likelihood of changing health behaviors. This is an interesting study in which a comprehensive dataset has been collected. The manuscript is clear, concise, and presented in a well-structured manner. The introduction is relevant, and sufficient information about the previous study findings is presented for readers to follow the present study rationale and procedures. Overall, the results are clear and compelling. The cited references are current, mainly within the last 1-2 years.

Several changes are required before this manuscript can be considered for publication in the International Journal of Environmental Research and Public Health. Please find below my comments which I believe would help improve the quality of the manuscript.

  1. Names of the COVID-19 vaccine that people in the study declared to have been vaccinated are inconsistent throughout the manuscript. The Materials and Methods section lists the names of companies producing vaccines against COVID-19: Sinovac-Biotech and BioNTech-Fosun Pharma, while the Results section contains the name BioN-Tech vaccine. The names should be uniform throughout the text of the manuscript. It is appropriate to mention the name of the preparation, not just the name of the company producing the preparation. Moreover, for BNT162b2 (BioNTech-Fosun Pharma) vaccine, it is worth mentioning that it is an mRNA vaccine, and CoronaVac (Sinovac-Biotech) vaccine, that it is an inactivated pathogen vaccine.

Thank you for these helpful comments. We have updated our manuscript to keep the terms of each vaccine consistent throughout. We have also more clearly introduced each vaccine type in the Introduction including the name of the preparation, company producing the vaccine, and the vaccine type. After introducing each vaccine, we now consistently use the terms Comirnaty and CoronaVac to refer to each vaccine throughout the manuscript, as these are the common names used in Hong Kong to identify each vaccine:

“In May 2021, the Hong Kong government’s “vaccination for all” campaign began offering two types of COVID-19 vaccines to all adults [5]. The first, BNT162b2 (Comirnaty), produced by Fosun-BioNTech, is a mRNA vaccine, while the second, CoronaVac, produced by Sinovac-Biotech, is an inactivated virus vaccine [6].” (lines 42-46)

Reference:

  1. The Government of the Hong Kong Special Administrative Region “Early Vaccination for All” Campaign Launched Available online: https://www.info.gov.hk/gia/general/202105/31/P2021053100749.htm (accessed on 3 November 2021).
  2. The Government of the Hong Kong Special Administrative Region About the Vaccines Available online: https://www.covidvaccine.gov.hk/en/vaccine (accessed on 3 May 2022).

  1. Have studies conducted after vaccination with a different type of vaccine, e.g., mRNA-, adenovirus-, protein-based or whole virus vaccine, could alter the results presented in the manuscript. Do the authors suppose that the perception of the safety of using non-mRNA-based or inactivated pathogen vaccines is similar?

Thank you for your very interesting question. Throughout vaccine availability in Hong Kong, two types of vaccines have been offered: 1) a mRNA vaccine, BNT162b2 (Comirnaty), produced by Fosun-BioNTech and 2) an inactivated virus vaccine, CoronaVac, produced by Sinovac-Biotech. Residents in Hong Kong have always been allowed to choose which vaccine they receive. While there is no literature we can point to in Hong Kong on determinants of vaccine-type preference, overall Comirnaty has been preferred over CoronaVac, though CoronaVac has been more popular in older aged adults and those with mainland Chinese nationality. At the end of our data collection time, August 29, 2021, 2.5 million first doses of Comirnaty had been administered while about 1.5 million doses of CoronaVac had been administered. This difference has diminished over time as older adults are vaccinated more. As of May 2022, first doses of Comirnaty and CoronaVac stand at 3.7 and 2.9 million, respectively. Generally, both vaccines are seen as safe, though people think Comirnaty results in more side effects but provides a stronger protection while CoronaVac would have more mild side effects but also more mild protection, which has been found in the two studies listed below (Mok et al., 2022; Yiu et al., 2022). Though predominately ethnically Chinese, in a multi-ethnic and multi-national city like Hong Kong, the place of origin of vaccine has also been important. Non-Chinese and younger people tend to prefer Comirnaty while older adults and those more recently coming from mainland China prefer CoronaVac. Unfortunately, we did not ask our participants the reasons for vaccine-type preference, nor did we ask those unvaccinated participants if they had a vaccine-type preference.

Your question brought up two important points for us to address. First, we think the limitations section should acknowledge that the time of data collection may affect the vaccination rate as it was still somewhat early in the vaccine availability period and it was not until later in the year that more evidence emerged on vaccine safety and efficacy. Second, we want to acknowledge that specific concerns to vaccine-type were not considered and that these concerns specifically may also affect vaccine uptake. These two added limitation now state:

“Fifth, our data was collected at a somewhat early stage of vaccine availability in Hong Kong. After our data collection, further evidence emerged addressing the immunogenicity of the two vaccines available in Hong Kong [44,45]. Such reports may have an effect on vaccine uptake, and specifically, type of vaccine taken. Additionally, the type of vaccines available in Hong Kong may also have an effect on vaccine uptake, though we did not ask those unvaccinated about vaccine-type preference and were not able to control for this in our analysis. However, in Hong Kong both a German developed mRNA vaccine, Comirnaty, and a Chinese developed inactivated virus vaccine, CoronaVac, were both equally accessible options to the population; therefore, concerns related to type of and location of vaccine should be minimal.” (lines 398-407)

References:

  1. Mok, C.K.P.; Cohen, C.A.; Cheng, S.M.S.; Chen, C.; Kwok, K.O.; Yiu, K.; Chan, T.O.; Bull, M.; Ling, K.C.; Dai, Z.; et al. Comparison of the Immunogenicity of BNT162b2 and CoronaVac COVID-19 Vaccines in Hong Kong. Respirology 2022, 27, 301–310, doi:10.1111/RESP.14191.
  2. Cheng, S.M.S.; Mok, C.K.P.; Leung, Y.W.Y.; Ng, S.S.; Chan, K.C.K.; Ko, F.W.; Chen, C.; Yiu, K.; Lam, B.H.S.; Lau, E.H.Y.; et al. Neutralizing Antibodies against the SARS-CoV-2 Omicron Variant BA.1 Following Homologous and Heterologous CoronaVac or BNT162b2 Vaccination. Nat. Med. 2022 283 2022, 28, 486–489, doi:10.1038/s41591-022-01704-7.

  1. Apart from those listed in the manuscript's discussion section, what are the study's limitations?

Aside from the limitation added in response to the question above, we have taken into consideration further limitations present in our study and have expanded the limitations section to include these. First, we included the potential for social-desirability bias that may exist in self-reported data, when respondents answer how they think would be more socially desirable rather than their actual experience. While this is a real concern in self-reported data, we are not aware of any prior literature that would point to this being of special concern within our study population. We have also acknowledged that our study did not use an internationally standardized vaccine attitudes instrument, making our findings not directly comparable with previous studies, and that our study’s measurements require future validation. We have also added a few sentences to discuss the direction of future research.

“Second, as our survey relies on self-reported information, our study is subject to recall bias and social-desirability bias, where respondents answer what they believe to be socially desirable rather than their actual experience. However, since the events in the survey took place recently, recall bias should be minimized, while we have no prior indication that social-desirability bias would be of significant concern within our study population.” (lines 385-390)

“Finally, our questionnaire was developed for exploratory purposes for this unique population and no internationally standardized vaccine attitudes instrument was available, making our findings not directly comparable with previous studies. Future studies are needed for the FDW populations outside of Hong Kong in countries and regions with different societal and policy structures that could influence vaccine uptake differently. A longitudinal study would also help in addressing the causal relationship between variables found in this study to be associated with vaccine uptake. Qualitative studies may also help to understand in more depth the barriers and mechanisms to vaccine uptake within this community.” (lines 407-415)

Minor comments

  1. Notation of numbers are misspelled throughout the text of the manuscript. Decimals should be written: 5.7 instead of 5∙7 (Line 164).

Thank you for pointing this out. We have made changes throughout the manuscript to adjust the location of the decimal place.

  1. Please reformat the Tables to improve their readability. There should be a space between the footnotes Table1 and Table2 and the description of the tables.

We appreciate the consideration of the readability of the tables. We have created a space between the footnotes in Table 1 and Table 2 and the manuscript text. As the layout of the manuscript has been changed from our initial submission, we will be sure to clarify the layout of the tables with the journal prior to any publication to ensure their readability.

  1. Grammatical errors and typos should be corrected throughout the manuscript.

Thank you for your careful reading of the manuscript. We have gone through it more carefully now to fix grammatical errors and typos and improve overall readability.

Given these shortcomings, the manuscript requires minor revisions.

We thank the reviewer for all of their comments and suggestions and have made changes according to their comments.

Round 2

Reviewer 1 Report

Thank you for the revisions